

# Are dietary factors associated with cardiometabolic risk factors in patients with non-alcoholic fatty liver disease?

Gulsum Gizem Topal[1], Sumeyra Sevim[2], Damla Gumus[3], Hatice Yasemin Balaban[4], Muşturay Karçaaltıncaba[5] and Mevlude Kizil[3]

[1] Department of Nutrition and Dietetics, Faculty of Health Sciences, Akdeniz University, Antalya, Turkey
[2] Department of Nutrition and Dietetics, Faculty of Health Sciences, Ankara Medipol University, Ankara, Turkey
[3] Department of Nutrition and Dietetics, Faculty of Health Sciences, Hacettepe University, Ankara, Turkey
[4] Department of Gastroenterology, Faculty of Medicine, Hacettepe University, Ankara, Turkey
[5] Department of Radiology, Faculty of Medicine, Hacettepe University, Ankara, Turkey

Corresponding author
Mevlude Kizil,
mkizil@hacettepe.edu.tr

## ABSTRACT

**Background:** Non-alcoholic fatty liver disease (NAFLD) is intricately linked with dietary patterns and metabolic homeostasis. Therefore, the present study focused to investigate the relation between dietary patterns and cardiometabolic risk factors related to fatty liver in NAFLD patients.

**Methods:** This cross-sectional study included 117 individuals whose body mass index (BMI) threshold of 25 or above diagnosed with NAFLD by magnetic resonance imaging. The hospital database was used to review the patients' medical records such as lipid parameters, and fasting blood sugar. Anthropometric measurements and body composition were measured by researchers. Likewise, data from 24-h dietary recalls of individuals were collected to analyze their energy and nutrient intakes besides calculating dietary insulin index (DII), dietary insulin load (DIL), dietary glycemic index (DGI), and dietary glycemic load (DGL).

**Results:** Participants consuming diets with distinct levels of DII, DIL, DGI, and DGL exhibited variations in dietary energy and nutrient intake. Specifically, differences were noted in carbohydrate intake across quartiles of DII, DIL, DGI, and DGL, while fructose consumption showed variability in DGL quartiles ($p \leq 0.05$). Moreover, sucrose intake demonstrated distinctions in both DII and DGL quartiles ($p \leq 0.05$). No statistical difference was found in biochemical parameters and the fatty liver index among different levels of DII, DIL, DGI, and DGL ($p > 0.05$). After adjusting for potential confounders, participants with a higher DGI had four times greater odds of developing metabolic syndrome compared to those in the bottom quartile (OR, 4.32; 95% CI [1.42–13.11]).

**Conclusion:** This study provides initial evidence of the intricate association between dietary factors and NAFLD, emphasizing the necessity for further research including prospective designs with larger sample sizes, to garner additional insights.

## INTRODUCTION

Non-alcoholic fatty liver disease (NAFLD) has emerged as a pervasive and complex health concern, intricately linked with dietary patterns and metabolic homeostasis (*Ren et al., 2023*). It covers a histological spectrum that includes simple steatosis, non-alcoholic steatohepatitis, fibrosis, and cirrhosis, with the possibility of progressing to liver failure and hepatocellular carcinoma (HCC) (*European Association for the Study of the Liver (EASL), European Association for the Study of Diabetes (EASD) & European Association for the Study of Obesity (EASO), 2016*). Recent reports estimate that the worldwide prevalence of NAFLD is around 38% (*Golabi, Owrangi & Younossi, 2024*; *Younossi et al., 2023*). Initially thought to be a disease primarily affecting Western countries, NAFLD now impacts roughly one-third of the global population across all regions. Interestingly, NAFLD is more frequently observed in regions such as South America, the Middle East and North Africa (MENA), Asia, and various other developing areas (*Younossi et al., 2023*). As the global prevalence of NAFLD continues to rise, it is imperative to elucidate the intricate factors contributing to its development and progression becomes important (*Le et al., 2022*). Among the multifaceted determinants, diet stands out as a pivotal modulator, wielding significant influence over the metabolic pathways that govern hepatic lipid accumulation (*Del Bo' et al., 2023*).

The dietary insulin index (DII) and dietary insulin load (DIL) represent dynamic parameters that reflect the postprandial insulin response to foods that consumed (*Behbahani et al., 2023*). The DII is defined as a direct measure of the postprandial insulin response to a test food, comparing it to an isoenergetic portion of a reference food (similar to the glycemic index, using either glucose or white bread). Unlike the glycemic index (GI), the DII is considered as more suitable for investigating the association with the development of chronic diseases because it is directly grounded in the insulin response. The DIL can be established by multiplying the DII of each food by its energy content and consumption frequency (*Mirmiran et al., 2015*; *Qi, Xia & Xu, 2023*). Results from epidemiological studies suggest a potential association between DIL and DII with certain metabolic disorders, such as heightened levels of inflammatory cytokines, atherogenic lipid profiles, and the onset of non-communicable chronic diseases (*Anjom-Shoae et al., 2020*; *Kazemi et al., 2021*; *Lambadiari, Korakas & Tsimihodimos, 2020*).

The relationship between DII, DIL, and NAFLD is a complex interplay that involves multiple physiological and metabolic pathways. Elevated insulin levels, often triggered by the consumption of high-glycemic-index foods, are related with increased hepatic lipogenesis. This elevated lipogenesis contributes to the accumulation of triglycerides in hepatocytes, a characteristic feature of NAFLD (*Smith et al., 2020*; *Softic et al., 2017*). Insulin resistance and hepatic lipid accumulation are intricately linked with systemic inflammation, a key player in the progression of NAFLD (*Petrescu et al., 2022*; *Smith et al., 2020*). Certain dietary patterns that promote elevated insulin levels may contribute to

chronic low-grade inflammation, further exacerbating liver damage and fostering an environment conducive to the development of more severe forms of NAFLD (*Valibeygi et al., 2023*). Moreover, previous research revealed that consuming diets characterized by elevated overall dietary glycemic index (DGI) and load (DGL) might amplify markers of systemic inflammation, recognized as pivotal elements contributing to the risk of NAFLD in individuals with T2DM (*Jayedi et al., 2022*; *Popov et al., 2022*).

Examining the association between dietary insulin dynamics and NAFLD holds implications for clinical management since understanding how specific dietary patterns influence insulin responses can inform dietary interventions aimed at mitigating the risk of NAFLD or improving outcomes in individuals already affected by the condition. As insulin is a regulator of lipid metabolism and hepatic glucose uptake, exploring the relationship between these dietary metrics and NAFLD holds promise for unravelling novel insights into the intricate interplay between nutrition and liver health. While existing literature has underscored the importance of dietary composition in NAFLD, a comprehensive understanding of the association between DII and DIL in NAFLD remains an evolving frontier. In contrast to previous studies that predominantly examined individual dietary factors, this research employs an integrative approach by incorporating multiple dietary indices. This methodology offers a more comprehensive perspective on the dietary influences affecting NAFLD. Hence, the present research aims to examine the association of DII, DIL, DGI and DGL with cardiometabolic risk factors and indices related to fatty liver in NAFLD patients.

## MATERIALS AND METHODS

### Patients

This cross-sectional study included 117 individuals aged between 18 and 65, who were diagnosed with NAFLD within the previous 6 months at gastroenterology clinics of Hacettepe University Hospital between December 2022 and September 2023. The diagnosis of NAFLD was based on clinical findings together with hepatic magnetic resonance imaging. A body mass index (BMI) threshold of 25 or above as an inclusion criterion was employed. Furthermore, individuals who smoked, consumed an extreme amount of alcohol (20 g/day), were breastfeeding or pregnant, took medications or supplements that could influence physiological functioning, or had a history of other liver problems, renal failure and cancer diagnosis were excluded from the study.

The hospital database was used to review the patients' medical records, and the last results for ALT, AST, GGT, lipid parameters, and fasting blood sugar were obtained. Fatty Liver Index (FLI) is a clinical tool used to predict the likelihood of NAFLD in individuals. The FLI was determined utilizing an algorithm that incorporates BMI, waist circumference, triglycerides, and GGT. An FLI value equal to or greater than 60 was employed as an indicator of hepatic steatosis (*Bedogni et al., 2006*).

The research was conducted in compliance with the principles outlined in the Declaration of Helsinki and received approval from the Hacettepe University Health Sciences Research Ethics Committee (GO 22/1045). The signed informed consent was obtained from each participant of the present study.

### Dietary and physical activity assessment

Data from 24-h dietary recalls of individuals were collected to analyze energy and nutrient intakes besides calculating DII, DIL, DGI and DGL by utilizing a computer-aided nutrition program (Nutrition Information System BeBIS 8.0 package program). The 24-h food consumption record obtained *via* the Food and Nutrition Photo Catalog was used to determine the energy and nutrient content of the foods consumed daily by individuals (*Rakicioğlu et al., 2010*). The International Physical Activity Questionnaire-Short Form was utilized to assess the physical activity status (*Craig et al., 2003*).

### Calculation of dietary insulin index and glycemic index

This study utilized glucose as a reference food to calculate the GI variable. The overall DGI was determined by adding the carbohydrate content (g) of each food, multiplying it by its GI, and then dividing it by the total carbohydrate intake (*Foster-Powell & Miller, 1995*). The total DGL was calculated by multiplying each food item's carbohydrate content by its GI value (*Zhang et al., 2006*).

The Food Insulin Index (FII) measures the rise in insulin levels over a 2-h period in response to a 1,000-kJ (239 kcal) serving of a test food, divided by the insulin response observed after consuming a 1,000-kJ (239 kcal) portion of a reference food. From earlier investigations, the insulin index values for different foods were collected (*Bell et al., 2016*; *Holt, Miller & Petocz, 1997*). Due to the extremely low levels of calories, carbohydrates, proteins, and fats found in tea, coffee, spices, and salt, these items were assigned an insulin index score of 0. The FIIs of equivalent foods were utilized for the other foods that were not included in the food lists of the earlier study. These FIIs were based on correlations between the foods' energy, fiber, carbohydrate, protein, and fat contents. Using the following formula, the insulin load of each food was initially calculated to determine DILs: insulin index of a specific food multiplied by the food energy content per 1 g of that food, and then multiplied by the quantity of that food (g/d). The overall insulin index was determined by division of IL by the daily caloric intake (*Arabshahi et al., 2023*).

$$DIL = \sum_{a=1}^{n} Insulin\ index\ of\ food\ a \times Energy\ of\ food\ a \times Quantity\ of\ a$$

### Anthropometric and body composition assessment

Body weight and body composition (body fat, muscle, and fluid mass) were measured using a TANITA brand MC180 model bioelectrical impedance analysis (BIA) device. The height of the individuals was measured using a stadiometer, barefoot, and with the head in the Frankfort plane. The BMI was computed by dividing the body weight (kg) by the square of height ($m^2$) (*World Health Organization WHO, 2000*). Waist circumference was measured with individuals standing in an upright position, parallel to the ground, using a rigid tape measure from the midpoint between the lowest rib bone and the crista iliac bone.

## Cardiometabolic risk factors definition

Obesity, dyslipidemia, Metabolic Syndrome (MetS), and cardiovascular diseases (CVD) were considered as cardiometabolic risk factors in this study. Individuals with a BMI >30 kg/m$^2$ were considered to have obesity (*World Health Organization WHO, 2000*). Dyslipidemia was characterized by elevated serum levels, including total cholesterol ≥200 mg/dL, low-density cholesterol (LDL-C) ≥100 mg/dL, triglycerides (TG) ≥150 mg/dL, and high-density lipoprotein cholesterol (HDL-C) for female <50 mg/dl and for male <40 mg/dl, or a documented history of dyslipidemia medication use (*National Cholesterol Education Program (NCEP) Expert Panel on Detection, Evaluation, and Treatment of High Blood Cholesterol in Adults (Adult Treatment Panel III), 2002*). Furthermore, the diagnostic criteria of the International Diabetes Federation were employed to identify individuals with MetS. Individuals with a medical history and/or ongoing treatment for one or more heart-related conditions were categorized as individuals with CVDs (*Day, 2007*).

## Statistical analysis

SPSS 23 software was used for all statistical analyses. Mean, median, standard deviation and lower and upper values were used for descriptive analyses. Spearman correlation coefficient was used to examine the degree and direction of the relationship between the variables. To ascertain the relationship between obesity, dyslipidemia, MetS, CVD and DII, DIL, DGI, DGL scores in the fatty liver population, logistic regression model analysis was conducted. In all models, participants in the first quartiles of DII, DIL, DGI, and DGL were considered as the reference group. Tukey's multiple comparison test was applied to determine statistically significant differences between groups. Values with $p \leq 0.05$ with 95% confidence intervals (CIs) were considered statistically significant.

# RESULTS

## Demographic characteristics of the participants

The study population comprised 56 men and 61 women. Table 1 shows the demographic characteristics of individuals distributed across quartiles of DII (Q1; 16.05–34.73, Q2; 34.89–41.65, Q3; 41.86–49.55, Q4; 49.70–69.01), DIL (Q1; 12395.39–155328.38, Q2; 163285.85–257751.76, Q3; 259840.92–362815.22, Q4; 374115.85–784256.20), DGI (Q1; 28.55–46.98, Q2; 48.68–57.25, Q3; 57.33–63.75, Q4; 64.06–252.63), and DGL (Q1; 11.62–54.47, Q2; 56.93–85.36, Q3; 86.69–135.78, Q4; 138.08–608.58). The demographic characteristics were not statistically different across DII quartiles. Moreover, no significant differences were observed in parameters including age, physical activity level, and BMI across DII, DIL, DGI and DGL quartiles. On the other hand, a significant observation was the predominant male representation among participants with higher DIL, DGI and DGL levels ($p = 0.001$).

Other significant differences were noted in body fat percentages across the quartiles of both DGI and DGL. Participants in the highest quartiles of DGI and DGL exhibited

**Table 1  Demographic characteristics of the participants in each quartile of DII, DIL, DGI, and DGL.**

| | DII | | | | | DIL | | | | | DGI | | | | | DGL | | | | | Total |
|---|---|---|---|---|---|---|---|---|---|---|---|---|---|---|---|---|---|---|---|---|---|
| | Q1 | Q2 | Q3 | Q4 | P1 / *P2 | Q1 | Q2 | Q3 | Q4 | P1 / *P2 | Q1 | Q2 | Q3 | Q4 | P1 / *P2 | Q1 | Q2 | Q3 | Q4 | P1 / *P2 | Total |
| Frequency, n | 29 | 30 | 29 | 29 | | 29 | 30 | 29 | 29 | | 29 | 30 | 29 | 29 | | 29 | 30 | 29 | 29 | | |
| Age | 45.9 ± 10.13 | 45.0 ± 11.47 | 47.4 ± 12.88 | 42.8 ± 12.47 | 0.398 | 47.0 ± 9.96 | 46.4 ± 11.35 | 46.5 ± 14.45 | 41.1 ± 10.19 | 0.073 | 47.0 ± 12.07 | 44.2 ± 11.69 | 45.3 ± 9.97 | 44.7 ± 13.43 | 0.843 | 48.0 ± 10.22 | 47.9 ± 11.63 | 44.2 ± 13.07 | 40.9 ± 10.97 | 0.060 | 45.1 ± 11.43 |
| **Gender** | | | | | | | | | | | | | | | | | | | | | |
| Men | 14 (48.3) | 12 (40.0) | 13 (44.8) | 17 (58.6) | 0.532 | 12 (41.4)$^{a}$ | 9 (31.0)$^{b}$ | 14 (46.7)$^{c}$ | 21 (72.4)$^{d}$ | 0.013 | 10 (34.5)$^{a}$ | 11 (36.7)$^{b}$ | 16 (55.2)$^{c}$ | 19 (65.6)$^{c}$ | 0.050 | 8 (27.6)$^{a}$ | 12 (40.0)$^{b}$ | 13 (44.8)$^{b}$ | 23 (79.3)$^{ab}$ | 0.001 | 56 (47.9) |
| Women | 15 (51.7) | 18 (60.0) | 16 (55.2) | 12 (41.4) | | 17 (58.6)$^{a}$ | 20 (69.0)$^{b}$ | 16 (53.3)$^{c}$ | 8 (27.6)$^{d}$ | | 19 (65.5)$^{a}$ | 19 (63.6)$^{b}$ | 13 (44.8)$^{c}$ | 10 (34.5)$^{c}$ | | 21 (72.4)$^{a}$ | 18 (60.0)$^{b}$ | 16 (55.2)$^{b}$ | 6 (20.7)$^{b}$ | | 61 (52.1) |
| **Physical activity level (IPAQ)** | | | | | | | | | | | | | | | | | | | | | |
| Inactive | 15 (51.7) | 22 (73.3) | 21 (72.4) | 22 (75.9) | 0.170 | 19 (65.6) | 22 (75.9) | 20 (66.7) | 19 (65.5) | 0.799 | 21 (72.4) | 21 (70.0) | 19 (65.5) | 19 (65.5) | 0.924 | 20 (69.0) | 22 (73.3) | 20 (69.0) | 18 (62.1) | 0.829 | 80 (68.4) |
| Minimal active | 14 (48.3) | 8 (26.7) | 8 (27.6) | 7 (24.1) | *0.835 | 10 (34.5) | 7 (24.1) | 10 (33.3) | 10 (34.5) | *0.790 | 8 (27.6) | 9 (30.0) | 10 (34.5) | 10 (34.5) | *0.773 | 9 (31.0) | 8 (26.7) | 9 (31.0) | 11 (37.9) | *0.927 | 37 (31.6) |
| Active | 0 | 0 | 0 | 0 | | 0 | 0 | 0 | 0 | | 0 | 0 | 0 | 0 | | 0 | 0 | 0 | 0 | | 0 |
| **Waist circumference (normal-above; ≥94 cm for men, and ≥80 cm for women)** | | | | | | | | | | | | | | | | | | | | | |
| Normal | 5 (17.2)$^{A}$ | 2 (6.9)$^{B}$ | 2 (6.9)$^{B}$ | 3 (10.3)$^{AB}$ | 0.510 / *0.017 | 2 (6.9)$^{A}$ | 3 (10.3)$^{A}$ | 2 (6.7)$^{A}$ | 5 (17.2)$^{B}$ | 0.510 / *0.017 | 1 (3.4)$^{aA}$ | 2 (6.7)$^{aA}$ | 2 (6.9)$^{aA}$ | 7 (24.1)$^{bB}$ | 0.040 / *0.031 | 1 (3.4) | 3 (10.1) | 2 (6.9) | 6 (20.7) | 0.155 / *0.102 | 12 (10.3) |
| Above | 24 (82.8)$^{A}$ | 28 (93.3)$^{B}$ | 27 (93.1)$^{B}$ | 26 (89.7)$^{AB}$ | | 27 (93.1)$^{A}$ | 26 (93.1)$^{A}$ | 28 (93.3)$^{A}$ | 24 (82.8)$^{B}$ | | 28 (96.6)$^{aA}$ | 28 (93.3)$^{aA}$ | 27 (93.1)$^{aA}$ | 22 (75.9)$^{bB}$ | | 28 (96.6) | 27 (90.0) | 27 (93.1) | 23 (79.3) | | 105 (89.7) |
| Body fat percentage (%) | 33.1 ± 8.88 | 33.1 ± 8.36 | 34.5 ± 8.36 | 32.1 ± 8.18 | 0.739 / *0.936 | 34.3 ± 8.13 | 34.6 ± 9.04 | 33.6 ± 7.44 | 30.3 ± 8.60 | 0.184 / *0.220 | 36.8 ± 6.04$^{a}$ | 34.0 ± 8.97$^{ab}$ | 32.1 ± 8.10$^{ab}$ | 29.7 ± 8.82$^{b}$ | 0.009 / *0.218 | 35.8 ± 6.74$^{a}$ | 33.3 ± 9.43$^{ab}$ | 33.9 ± 7.22$^{ab}$ | 29.7 ± 9.05$^{b}$ | 0.046 / *0.417 | 33.2 ± 8.39 |
| Body mass index-BMI (kg/m²) | 32.0 ± 6.27 | 31.6 ± 5.62 | 31.7 ± 4.82 | 31.8 ± 5.36 | 0.992 / *0.967 | 32.4 ± 6.08 | 32.3 ± 5.42 | 31.1 ± 4.80 | 31.4 ± 5.69 | 0.747 / *0.096 | 33.4 ± 5.27 | 31.5 ± 5.32 | 31.0 ± 5.05 | 31.1 ± 6.13 | 0.316 / *0.763 | 32.4 ± 4.82 | 30.9 ± 5.52 | 32.7 ± 5.60 | 31.1 ± 5.96 | 0.509 / *0.886 | 31.8 ± 5.47 |
| **Body mass index-BMI (kg/m²)** | | | | | | | | | | | | | | | | | | | | | |
| 25.0–29.9 kg/m² | 10 (34.5) | 11 (36.7) | 6 (20.7) | 10 (34.5) | 0.812 / 0.570 | 7 (24.1) | 10 (34.5) | 14 (46.7) | 6 (20.7) | 0.447 / 0.548 | 4 (13.8) | 12 (40.0) | 11 (37.9) | 10 (34.5) | 0.111 / *0.522 | 8 (27.6) | 14 (46.7) | 9 (31.0) | 6 (20.7) | 0.458 / 0.854 | 47 (40.2) |
| 30.0–34.9 kg/m² | 9 (31.0) | 10 (33.3) | 13 (34.5) | 9 (31.0) | | 12 (41.4) | 8 (27.6) | 10 (33.3) | 11 (37.9) | | 17 (58.6) | 7 (23.3) | 8 (27.6) | 9 (31.0) | | 11 (37.9) | 9 (30.0) | 11 (37.9) | 10 (34.5) | | 41 (35) |
| ≥35.0 kg/m² | 7 (24.1) | 6 (20.2) | 8 (27.6) | 8 (27.6) | | 26 (89.7) | 9 (31.0) | 5 (16.7) | 8 (27.6) | | 7 (24.1) | 9 (30.0) | 7 (24.1) | 6 (20.7) | | 9 (31.0) | 4 (13.3) | 8 (27.6) | 8 (27.6) | | 29 (24.8) |

**Notes:**
P1; ANOVA (Tukey's). Kruskal Wallis. chi-square. Bonferroni adjustment. Results with different letters (a–d) in the same row are significantly different.
* P2; ANCOVA, Pearson chi-square; age and gender as covariates.

**Table 2  Biochemical parameters of the participants in each quartile of DII and DIL.**

| | DII | | | | | DIL | | | | | Total |
|---|---|---|---|---|---|---|---|---|---|---|---|
| | Q1 | Q2 | Q3 | Q4 | P1 *P2 | Q1 | Q2 | Q3 | Q4 | P1 *P2 | |
| ALT (alanine transaminase) | 42.2 ± 22.62 | 57.5 ± 35.09 | 53.1 ± 51.71 | 45.7 ± 22.51 | 0.364 *0.781 | 48.6 ± 26.27 | 41.6 ± 21.61 | 43.6 ± 21.82 | 65.3 ± 55.73 | 0.253 *0.787 | 49.7 ± 35.15 |
| AST (aspartate transaminase) | 30.6 ± 11.89 | 37.6 ± 18.75 | 34.0 ± 18.67 | 30.2 ± 11.13 | 0.442 *0.843 | 34.3 ± 16.10 | 31.3 ± 11.22 | 30.2 ± 9.63 | 36.9 ± 22.50 | 0.830 *0.686 | 33.1 ± 15.66 |
| GGT (gamma-glutamyl transferase) | 42.0 ± 24.37 | 47.6 ± 33.51 | 53.4 ± 35.44 | 49.5 ± 27.49 | 0.733 *0.792 | 44.4 ± 25.35 | 43.2 ± 28.00 | 49.5 ± 33.45 | 55.3 ± 34.02 | 0.449 *0.954 | 48.1 ± 30.44 |
| HDL-C | 48.4 ± 11.96 | 46.6 ± 11.17 | 48.2 ± 11.46 | 46.6 ± 12.35 | 0.884 *0.494 | 45.9 ± 11.30 | 50.1 ± 13.13 | 48.7 ± 11.40 | 44.9 ± 10.35 | 0.197 *0.540 | 47.4 ± 11.62 |
| LDL-C | 131.3 ± 36.70 | 127.1 ± 29.74 | 133.0 ± 33.12 | 132.6 ± 28.24 | 0.710 *0.411 | 132.4 ± 34.06 | 128.6 ± 29.31 | 135.8 ± 35.99 | 126.9 ± 27.73 | 0.534 *0.845 | 130.9 ± 31.77 |
| Triglycerides | 164.0 ± 85.42 | 157.2 ± 69.90 | 157.5 ± 68.72 | 176.1 ± 75.76 | 0.689 *0.053 | 185.6 ± 79.54 | 136.2 ± 60.42 | 163.3 ± 59.70 | 169.6 ± 89.87 | 0.125 *0.125 | 163.7 ± 74.61 |
| Total cholesterol | 193.9 ± 45.33 | 190.0 ± 38.75 | 196.2 ± 44.21 | 187.1 ± 3594 | 0.839 *0.705 | 195.4 ± 38.99 | 192.8 ± 41.42 | 193.3 ± 47.89 | 185.5 ± 38.10 | 0.813 *0.994 | 191.8 ± 40.83 |
| Fasting blood glucose | 108.8 ± 29.25 | 112.7 ± 39.46 | 112.8 ± 39.13 | 104.2 ± 22.52 | 0.901 *0.224 | 108.8 ± 21.10 | 116.3 ± 41.42 | 112.6 ± 35.57 | 100.8 ± 30.80 | 0.197 *0.566 | 109.64 ± 33.17 |
| Fatty liver index (FLI) | 74.6 ± 25.64 | 77.2 ± 18.53 | 79.7 ± 17.62 | 76.8 ± 20.31 | 0.945 *0.782 | 79.7 ± 19.44 | 75.6 ± 22.70 | 75.3 ± 20.00 | 77.6 ± 20.71 | 0.691 *0.595 | 77.0 ± 20.55 |
| Fatty liver index (FLI) | | | | | | | | | | | |
| <60 | 7 (24.1) | 5 (16.7) | 4 (13.8) | 6 (20.7) | 0.759 *0.261 | 4 (13.8) | 6 (20.7) | 6 (20.0) | 6 (20.7) | 0.887 *0.364 | 22 (18.8) |
| ≥60 | 22 (75.9) | 25 (83.3) | 25 (86.2) | 23 (79.3) | | 25 (86.2) | 23 (79.3) | 24 (80.0) | 23 (79.3) | | 95 (1.2) |

**Notes:**
P1; ANOVA, Kruskal Wallis, Chi-square.
* P2; ANCOVA, Pearson chi-square; Age and gender as covariates.

**Table 3 Biochemical parameters of the participants in each quartile of DGI and DGL.**

| | DGI | | | | | DGL | | | | |
|---|---|---|---|---|---|---|---|---|---|---|
| | Q1 | Q2 | Q3 | Q4 | P1 / *P2 | Q1 | Q2 | Q3 | Q4 | P1 / *P2 |
| ALT (alanine transaminase) | 44.3 ± 34.00 | 44.4 ± 24.06 | 54.7 ± 29.08 | 55.7 ± 48.93 | 0.209 / *0.862 | 42.1 ± 24.99 | 45.6 ± 21.24 | 52.2 ± 33.76 | 59.0 ± 52.09 | 0.414 / *0.975 |
| AST (aspartate transaminase) | 32.2 ± 19.04 | 30.2 ± 11.51 | 33.3 ± 12.44 | 36.9 ± 18.34 | 0.368 / *0.508 | 32.1 ± 15.94 | 32.7 ± 10.09 | 35.4 ± 17.90 | 32.3 ± 18.12 | 0.681 / *0.666 |
| GGT (gamma-glutamyl transferase) | 42.3 ± 29.90 | 47.4 ± 27.54 | 54.9 ± 34.33 | 47.9 ± 29.94 | 0.383 / *0.622 | 42.1 ± 21.41 | 49.5 ± 36.51 | 52.8 ± 33.61 | 48.1 ± 28.42 | 0.800 / *0.487 |
| HDL-C | 49.1 ± 11.70 | 49.3 ± 12.05 | 45.0 ± 10.26 | 46.2 ± 12.36 | 0.267 / *0.726 | 51.0 ± 13.97 | 46.4 ± 9.85 | 47.1 ± 11.45 | 45.2 ± 10.60 | 0.316 / *0.392 |
| LDL-C | 132.8 ± 37.28 | 133.9 ± 29.39 | 132.3 ± 28.81 | 124.8 ± 31.81 | 0.613 / *0.978 | 138.3 ± 33.07 | 127.8 ± 32.50 | 135.7 ± 29.53 | 122.1 ± 30.82 | 0.193 / *0.443 |
| Triglycerides | 172.2 ± 88.02 | 145.9 ± 64.49 | 173.9 ± 82.26 | 163.3 ± 60.96 | 0.570 / *0.561 | 164.9 ± 75.44 | 147.7 ± 64.48 | 192.7 ± 88.98 | 149.9 ± 61.65 | 0.164 / *0.467 |
| Total cholesterol | 204.1 ± 47.31 | 189.6 ± 43.03 | 190.5 ± 37.10 | 183.0 ± 33.63 | 0.253 / *0.706 | 201.7 ± 44.11 | 190.2 ± 44.52 | 192.8 ± 38.15 | 182.4 ± 35.45 | 0.350 / *0.892 |
| Fasting blood glucose | 112.6 ± 33.06 | 114.3 ± 42.74 | 108.3 ± 30.33 | 103.2 ± 23.99 | 0.739 / *0.202 | 115.1 ± 41.12 | 110.2 ± 21.76 | 113.7 ± 36.92 | 99.6 ± 29.33 | 0.114 / *0.311 |
| Fatty liver index (FLI) | 78.9 ± 22.11 | 77.0 ± 21.52 | 76.7 ± 20.80 | 75.5 ± 18.49 | 0.654 / *0.999 | 77.4 ± 18.83 | 74.6 ± 22.39 | 79.8 ± 21.15 | 76.4 ± 20.41 | 0.501 / *0.725 |
| Fatty liver index (FLI) | | | | | | | | | | |
| <60 | 5 (17.2) | 4 (13.3) | 8 (27.6) | 5 (17.2) | 0.543 / *0.311 | 4 (13.8) | 6 (20.0) | 6 (20.7) | 6 (20.7) | 0.887 / *0.331 |
| ≥60 | 24 (82.8) | 26 (86.7) | 21 (72.4) | 24 (82.8) | | 25 (86.2) | 24 (80.0) | 23 (79.3) | 23 (79.3) | |

**Notes:**
P1; ANOVA, Kruskal Wallis, Chi-square.
* P2; ANCOVA, Pearson chi-square; Age and gender as covariates.

statistically lower body fat percentages ($p \leq 0.05$). Additionally, normal waist circumference in the highest quartile of DGI was statistically higher than in the lower quartile ($p = 0.040$). Moreover, there are significant differences in waist circumference across DII, DIL, and DGI according to adjusted by age and gender ($p \leq 0.05$). To ensure a comprehensive analysis, the results were examined separately for men and women. Only the body fat percentage (%) was significantly different across quartiles of DIL in men (see Table S1).

## Biochemical parameters of the participants

Tables 2 and 3 display the biochemical parameters across the quartiles of DII, DIL, DGI, and DGL. No significant difference was found not only in the biochemical parameters, and indices related to fatty liver (serum ALT, AST, GGT, HDL-C, LDL-C, triglycerides, total cholesterol, fasting blood glucose, and FLI) across the quartiles of DII, DIL, DGI, and DGL but also adjusted by age and gender. Moreover, the results were examined separately for men and women. Some significant differences were found in the biochemical parameters across quartiles of DII, DIL, DGI, and DGL in men and women (see Tables S2 and S3). Men in the third quartiles of DGL exhibited statistically higher triglycerides level ($p < 0.001$). Additionally, total cholesterol in the lowest quartile of DGI was statistically higher than in the highest quartile ($p = 0.006$) (see Table S3).

## Dietary energy and nutrient intake of the participants

Tables 4 and 5 display the dietary intakes of chosen foods and nutrients across the quartiles of DII, DIL, DGI, and DGL. The percentages of total fat intake in the highest quartile of DII were statistically lower than in the lower quartile whereas percentages of carbohydrate intake in the highest quartile of DII were statistically higher than in the lower quartiles ($p = 0.009$) but the significant was not observed by adjusted age and gender ($p > 0.05$). On contrary, percentages of sucrose in the lowest DII quartile were statistically higher than in the second quartile ($p = 0.045$), with the third and fourth quartiles of DII being higher than the others, though not significantly different. Regarding DIL, there were significant differences in total energy, percentages of carbohydrate intake, fiber, soluble and insoluble fiber, and percentages of saturated fatty acids consumption across the quartiles. The percentages of carbohydrate intake and total energy intake in the highest quartile of DIL were higher than the other quartiles. Similarly, total fiber, soluble, and insoluble fiber consumption in the highest quartile of DIL were statistically higher than in the second quartile ($p \leq 0.05$). In contrast, saturated fatty acids in the second quartile were statistically higher than in the other quartiles ($p < 0.001$). However, in Table 4 shows that the n-6:n-3 ratio, which can be considered a risk factor for cardiovascular diseases, is statistically significantly lower in the first quartile of DIL compared to the third and fourth quartiles ($p = 0.013$). When the statistical analyzes were adjusted by age and gender, there are significant differences in only total energy, fiber, and soluble fiber across DIL quartiles ($p \leq 0.05$).

In the present study, total energy intake in the lowest DGI quartile was statistically lower than in the third and fourth quartiles, given that the percentages of protein intake in the

**Table 4 Participants' dietary energy and nutrient intake across quartiles of DII and DIL.**

| Nutrients | DII | | | | | DIL | | | | | Total |
|---|---|---|---|---|---|---|---|---|---|---|---|
| | Q1 | Q2 | Q3 | Q4 | P1 *P2 | Q1 | Q2 | Q3 | Q4 | P1 *P2 | |
| Energy (kcal/d) | 1,976.0 ± 1,000.48 | 1,707.0 ± 597.11 | 1,808.6 ± 921.14 | 1,754.8 ± 601.12 | 0.598 *0.690 | 1,555.9 ± 927.06[aA] | 1,453.0 ± 480.21[bB] | 1,712.7 ± 386.15[C] | 2,524.5 ± 800.53[dD] | 0.000 *0.000 | 1,810.7 ± 795.72 |
| Protein (% energy) | 17.3 ± 6.12 | 16.8 ± 4.90 | 15.4 ± 5.15 | 14.7 ± 4.67 | 0.256 *0.605 | 16.1 ± 7.17 | 16.8 ± 4.90 | 15.8 ± 4.26 | 15.7 ± 4.48 | 0.722 *0.062 | 16.1 ± 5.27 |
| Total fat (% energy) | 35.6 ± 11.72[ab] | 39.5 ± 9.20[a] | 32.2 ± 11.54[b] | 30.7 ± 9.42[b] | 0.009 *0.518 | 34.9 ± 13.47 | 37.6 ± 11.84 | 33.6 ± 9.82 | 32.2 ± 7.56 | 0.286 *0.482 | 35.5 ± 10.93 |
| Carbohydrate (% energy) | 41.8 ± 11.34[aA] | 41.7 ± 10.27[aA] | 47.0 ± 12.83[abAB] | 50.0 ± 9.35[bB] | 0.009 *0.004 | 40.7 ± 13.80[a] | 42.2 ± 8.85[ab] | 47.8 ± 10.49[b] | 49.6 ± 10.10[b] | 0.005 *0.675 | 45.1 ± 11.44 |
| Fiber (g/d) | 26.2 ± 14.95 | 21.1 ± 9.69 | 25.9 ± 15.15 | 20.0 ± 10.00 | 0.145 *0.349 | 21.4 ± 15.88[abAB] | 18.6 ± 8.30[aA] | 23.5 ± 8.64[abB] | 29.7 ± 14.66[bB] | 0.007 *0.004 | 23.3 ± 12.83 |
| Soluble fiber (g/d) | 9.8 ± 7.28 | 6.8 ± 4.11 | 8.0 ± 5.55 | 6.5 ± 3.78 | 0.387 *0.146 | 9.0 ± 8.02[abA] | 5.3 ± 2.69[aB] | 7.1 ± 3.37[abB] | 9.6 ± 5.24[bA] | 0.001 *0.046 | 7.8 ± 5.44 |
| Insoluble fiber (g/d) | 16.3 ± 11.22 | 13.4 ± 7.17 | 16.5 ± 10.90 | 13.5 ± 7.02 | 0.246 *0.729 | 12.1 ± 10.90[b] | 12.7 ± 6.10[b] | 15.0 ± 6.67[ab] | 19.8 ± 10.81[a] | 0.005 *0.146 | 14.9 ± 9.28 |
| Fructose (g/d) | 13.3 ± 15.41 | 13.1 ± 15.38 | 15.4 ± 10.62 | 14.5 ± 10.36 | 0.398 *0.875 | 13.4 ± 14.60 | 11.2 ± 8.69 | 12.8 ± 11.25 | 18.9 ± 15.86 | 0.078 *0.712 | 14.1 ± 13.06 |
| Sucrose (% total energy) | 5.2 ± 5.67[a] | 4.9 ± 3.29[b] | 6.4 ± 3.98[ab] | 9.7 ± 9.03[ab] | 0.045 *0.067 | 6.7 ± 8.95 | 5.3 ± 2.86 | 7.1 ± 6.35 | 7.0 ± 4.82 | 0.280 *0.678 | 6.5 ± 6.13 |
| Saturated fatty acids (% total energy) | 10.8 ± 5.20 | 13.7 ± 4.79 | 13.4 ± 5.19 | 14.2 ± 9.04 | 0.229 *0.300 | 13.6 ± 10.93[a] | 14.0 ± 4.23[b] | 12.6 ± 3.77[c] | 11.9 ± 3.45[ac] | 0.000 *0.435 | 13.0 ± 6.34 |
| Mono-unsaturated fatty acids (% total energy) | 12.2 ± 6.06 | 14.8 ± 5.17 | 13.8 ± 6.20 | 12.0 ± 6.79 | 0.062 *0.495 | 14.2 ± 10.60 | 14.3 ± 3.56 | 12.6 ± 3.64 | 11.8 ± 3.30 | 0.088 *0.669 | 13.2 ± 6.11 |
| Poly-unsaturated fatty acids (% total energy) | 7.2 ± 5.75 | 8.8 ± 4.12 | 7.2 ± 3.79 | 8.1 ± 8.82 | 0.063 *0.889 | 9.5 ± 9.86 | 7.9 ± 4.38 | 7.0 ± 3.82 | 7.1 ± 2.98 | 0.807 *0.269 | 7.9 ± 5.91 |
| Cholesterol (mg/d) | 277.9 ± 196.46 | 282.5 ± 160.12 | 324.2 ± 191.34 | 256.4 ± 198.52 | 0.574 *0.482 | 248.5 ± 216.33 | 283.5 ± 181.04 | 266.0 ± 161.22 | 343.6 ± 178.48 | 0.000 *0.435 | 285.2 ± 186.22 |
| n-6: n-3 | 7.4 ± 6.81 | 12.4 ± 9.54 | 9.6 ± 7.08 | 9.3 ± 9.78 | 0.099 *0.932 | 7.3 ± 9.61[a] | 9.4 ± 6.74[ab] | 10.9 ± 8.62[b] | 11.1 ± 8.68[b] | 0.013 *0.294 | 9.7 ± 8.51 |

**Notes:**

P1; ANOVA. Kruskal Wallis. Tukey's. Results with different letters (a–d) in the same row are significantly different.

* P2; ANCOVA, age and gender as covariates. Results with different letters (A–D) in the same row are significantly different.

**Table 5 Participants' dietary energy and nutrient intake across quartiles of DGI and DGL.**

| Nutrients | DGI | | | | | DGL | | | | |
|---|---|---|---|---|---|---|---|---|---|---|
| | Q1 | Q2 | Q3 | Q4 | P1 *P2 | Q1 | Q2 | Q3 | Q4 | P1 *P2 |
| Energy (kcal/d) | 1,400.0 ± 531.84[aA] | 1,821.8 ± 817.64[abAB] | 2,062.0 ± 900.68[BB] | 1,958.7 ± 757.43[bB] | 0.007 *0.048 | 1,244.5 ± 643.40[aA] | 1,814.3 ± 604.62[bB] | 1,741.4 ± 622.05[bB] | 2,442.4 ± 832.66[cC] | 0.000 *0.000 |
| Protein (% energy) | 18.6 ± 5.97[a] | 17.8 ± 4.36[a] | 14.5 ± 3.37[ab] | 13.3 ± 5.32[b] | 0.000 *0.000 | 18.4 ± 5.71[A] | 15.5 ± 4.85[B] | 15.3 ± 5.53[B] | 15.1 ± 4.45[B] | 0.053 *0.004 |
| Total fat (% energy) | 37.8 ± 10.95 | 35.9 ± 9.77 | 33.6 ± 11.23 | 30.9 ± 11.07 | 0.091 *0.416 | 38.9 ± 9.77[aA] | 35.9 ± 13.18[abA] | 32.9 ± 10.20[abB] | 30.4 ± 8.53[bB] | 0.017 *0.019 |
| Carbohydrate (% energy) | 39.8 ± 12.17[aA] | 44.7 ± 8.77[abB] | 51.5 ± 11.81[bC] | 44.4 ± 10.16[aB] | 0.001 *0.001 | 40.2 ± 10.91[aA] | 41.2 ± 10.01[aA] | 50.6 ± 11.02[bB] | 48.6 ± 10.62[bcB] | 0.000 *0.000 |
| Fiber (g/d) | 21.7 ± 10.21 | 22.1 ± 8.89 | 27.7 ± 18.70 | 21.7 ± 10.92 | 0.207 *0.795 | 17.8 ± 9.33[b] | 22.7 ± 6.93[b] | 24.5 ± 15.99[ab] | 28.0 ± 15.28[bc] | 0.010 *0.089 |
| Soluble fiber (g/d) | 6.8 ± 3.48 | 6.2 ± 3.57 | 9.7 ± 6.42 | 13.2 ± 6.95 | 0.202 *0.167 | 6.6 ± 6.41[a] | 7.2 ± 4.49[a] | 7.7 ± 5.04[ab] | 9.6 ± 5.51[b] | 0.008 *0.491 |
| Insoluble fiber (g/d) | 14.3 ± 7.45 | 14.3 ± 6.38 | 17.8 ± 14.08 | 13.2 ± 6.95 | 0.260 *0.742 | 10.7 ± 5.58[aA] | 14.0 ± 4.78[bB] | 16.5 ± 11.93[bBC] | 18.3 ± 11.20[cC] | 0.010 *0.040 |
| Fructose (g/d) | 14.7 ± 9.32 | 10.8 ± 9.27 | 17.0 ± 16.21 | 13.8 ± 15.70 | 0.172 *0.151 | 10.7 ± 7.23[a] | 9.3 ± 7.80[a] | 14.5 ± 10.33[ab] | 22.0 ± 19.54[b] | 0.002 *0.070 |
| Sucrose (% total energy) | 4.7 ± 2.91 | 5.3 ± 2.86 | 7.3 ± 6.58 | 8.9 ± 9.14 | 0.392 *0.210 | 4.3 ± 2.32[aA] | 4.2 ± 2.73[bA] | 9.9 ± 7.98[abB] | 7.9 ± 7.37[abB] | 0.002 *0.017 |
| Saturated fatty acids (% total energy) | 20.1 ± 10.84 | 26.0 ± 13.74 | 27.5 ± 12.29 | 24.2 ± 13.81 | 0.368 *0.516 | 13.3 ± 5.51[abc] | 13.0 ± 4.38[a] | 13.8 ± 8.82[b] | 11.9 ± 6.06[c] | 0.006 *0.610 |
| Mono-unsaturated fatty acids (% total energy) | 13.0 ± 4.84 | 14.0 ± 4.03 | 12.2 ± 6.32 | 13.6 ± 8.51 | 0.228 *0.233 | 13.9 ± 4.95 | 13.5 ± 4.94 | 13.5 ± 8.50 | 11.8 ± 8.65 | 0.110 *0.205 |
| Poly-unsaturated fatty acids (% total energy) | 7.3 ± 5.48 | 8.1 ± 3.83 | 6.8 ± 3.93 | 9.3 ± 8.96 | 0.340 *0.523 | 7.5 ± 5.33 | 7.4 ± 3.97 | 8.3 ± 7.89 | 8.3 ± 6.07 | 0.915 *0.949 |
| Cholesterol (mg/d) | 221.6 ± 176.40 | 292.6 ± 192.16 | 318.6 ± 163.97 | 307.7 ± 203.71 | 0.189 *0.241 | 208.0 ± 147.57[a] | 273.0 ± 176.60[ab] | 293.0 ± 197.35[ab] | 367.4 ± 193.00[b] | 0.011 *0.481 |
| n-6: n-3 | 8.7 ± 7.80 | 9.8 ± 8.04 | 10.3 ± 7.47 | 9.9 ± 10.71 | 0.324 *0.932 | 7.8 ± 7.86 | 10.2 ± 8.05 | 9.9 ± 7.31 | 10.8 ± 10.58 | 0.207 *0.880 |

**Notes:**
P1; ANOVA. Kruskal Wallis. Tukey's. Results with different letters (a–d) in the same row are significantly different.
* P2; ANCOVA, age and gender as covariates. Results with different letters (A–C) in the same row are significantly different.

highest quartile were lower than in the first and second quartiles. Conversely, the percentages of carbohydrate intake in the second and third quartiles were significantly higher than in the first and fourth quartiles of DGI ($p = 0.002$). When the statistical analyzes were adjusted by age and gender, there are significant differences in only total energy, and percentage of carbohydrate across DGI quartiles ($p \leq 0.05$). As seen in Table 5, significant differences were observed across the quartiles of DGL in various dietary and nutrient intakes ($p \leq 0.05$). In the highest DGL quartile, total energy, percentages of carbohydrates, total fiber, soluble and insoluble fiber intake, and fructose intake were significantly higher compared to the first quartile, while percentages of total fat in the lowest quartile were notably higher than in the highest quartile. Furthermore, percentages of sucrose intake in the lowest quartile were statistically higher than in the second quarter ($p = 0.002$), although the third and fourth quartiles exhibited higher values compared to the other quartiles ($p > 0.05$). Additionally, the percentages of saturated fatty acid in the third quartile were statistically higher than in the second and fourth quartiles of DGL ($p = 0.006$). On the other hand, as the statistical analyzes adjusted by age and gender, percentage of protein consumption was significantly different between DGL quartiles ($p = 0.004$).

The results were examined separately for men and women, and some differences were found in dietary energy and nutrient intake across quartiles of DII, DIL, DGI, and DGL (see Tables S4 and S5). In the highest DIL quartile, total energy, total fiber, and soluble fiber intake were significantly higher compared to the first quartile, while percentages of total fat and n-6:n-3 in the second quartile were notably higher than in the highest quartile in men (see Table S4). On the other hand, in the highest DIL quartile, total energy, percentages of sucrose intake were significantly higher compared to the other quartiles. Likewise, in the highest DII quartile, percentages of carbohydrate and sucrose intake were significantly higher compared to the lowest quartile (see Table S4).

In the third DGI quartile, percentage of carbohydrate intake were significantly higher compared to the first quartile, while percentages of sucrose in the third quartile were notably higher than in the lowest quartile in men (see Table S5). There were significant differences in total energy, percentage of protein, carbohydrate and mono-unsaturated fatty acids in both DGI and DGL quartiles. Moreover, the percentage of fat and sucrose and fructose (g/d) intake were different across DGL quartiles (see Table S5).

## Cardiometabolic risk factors of the participants

The odds of cardiometabolic risk factors across the quartiles of DII, DIL, DGI, and DGL are given in Table 6. Notably, individuals consuming a diet with a higher DGI had four times the odds of developing MetS compared to those with a lower DGI (OR, 4.32; 95% CI [1.42–13.11]). In the present study, associations were observed between DGI and obesity, with odds ratios for Q2 (OR, 0.22; 95% CI [0.06–0.81]), Q3 (OR, 0.23; 95% CI [0.06–0.85]), and Q4 (OR, 0.25; 95% CI [0.07–0.95]).

With the exception of the variables elucidated in this investigation, it was ascertained that DII, DIL, DGI, and DGL exhibited no statistically significant associations regarding
**Table 6 The odds of cardiometabolic risk factors across quartiles of DII, DIL, DGI, and DGL.**

| Quartiles | DII OR (95% CI) | | DIL OR (95% CI) | | DGI OR (95% CI) | | DGL OR (95% CI) | |
|---|---|---|---|---|---|---|---|---|
| | Crude | Adjusted* | Crude | Adjusted* | Crude | Adjusted* | Crude | Adjusted* |
| Obesity 1 | 1 | 1 | 1 | 1 | 1 | 1 | 1 | 1 |
| 2 | 0.93 [0.33–2.59] | 0.93 [0.33–2.59] | 0.75 [0.26–2.16] | 1.05 [0.34–3.18] | **0.24 [0.07–0.79]** | **0.24 [0.07–0.79]** | 0.34 [0.12–1.00] | 0.34 [0.12–1.00] |
| 3 | 2.13 [0.71–6.37] | 2.13 [0.71–6.42] | 0.53 [0.18–1.50] | 0.76 [0.26–2.22] | **0.22 [0.07–0.75]** | **0.22 [0.07–0.75]** | 0.86 [0.29–2.56] | 0.85 [0.28–2.55] |
| 4 | 1.15 [0.41–3.26] | 1.15 [0.40–3.28] | 1.00 [0.34–2.95] | 0.99 [0.97–1.02] | **0.22 [0.07–0.75]** | **0.22 [0.07–0.75]** | 0.74 [0.25–2.18] | 0.72 [0.24–2.17] |
| p trend | 0.618 | 0.620 | 0.606 | 0.595 | **0.016** | **0.016** | 0.470 | 0.462 |
| Dyslipidemia 1 | 1 | 1 | 1 | 1 | 1 | 1 | 1 | 1 |
| 2 | 0.68 [0.23–1.97] | 0.68 [0.23–1.98] | 0.32 [0.10–1.02] | 0.32 [0.10–1.02] | 1.41 [0.49–4.07] | 1.42 [0.49–4.08] | 0.68 [0.23–1.97] | 0.68 [0.23–1.98] |
| 3 | 0.64 [0.22–1.92] | 0.65 [0.22–1.92] | 0.52 [0.16–1.69] | 0.51 [0.15–1.71] | 2.22 [0.72–6.85] | 2.22 [0.72–6.85] | 1.41 [0.44–4.51] | 1.42 [0.44–4.51] |
| 4 | 1.72 [0.52–5.69] | 1.75 [0.53–5.82] | 0.50 [0.15–1.61] | 0.49 [0.14–1.64] | 1.34 [0.46–3.89] | 1.34 [0.46–3.89] | 0.74 [0.25–2.18] | 0.74 [0.25–2.22] |
| p trend | 0.419 | 0.424 | 0.192 | 0.192 | 0.426 | 0.425 | 0.537 | 0.540 |
| MetS 1 | 1 | 1 | 1 | 1 | 1 | 1 | 1 | 1 |
| 2 | 0.45 [0.16–1.30] | 0.44 [0.15–1.28] | 0.87 [0.31–2.46] | 0.83 [0.29–2.39] | 2.83 [0.98–8.12] | 2.81 [0.98–8.09] | 2.50 [0.86–7.27] | 2.47 [0.85–7.21] |
| 3 | 0.55 [0.19–1.62] | 0.52 [0.18–1.56] | 1.41 [0.49–4.07] | 1.32 [0.45–3.90] | **3.11 [1.06–9.08]** | **3.12 [1.07–9.15]** | 1.52 [0.54–4.28] | 1.50 [0.53–4.26] |
| 4 | 0.86 [0.29–2.56] | 0.81 [0.27–2.46] | 1.00 [0.35–2.84] | 0.93 [0.32–2.72] | **4.30 [1.42–13.00]** | **4.32 [1.42–13.11]** | 1.75 [0.62–4.99] | 1.68 [0.58–4.84] |
| p trend | 0.401 | 0.363 | 0.771 | 0.747 | **0.034** | **0.034** | 0.272 | 0.291 |
| CVD 1 | 1 | 1 | 1 | 1 | 1 | 1 | 1 | 1 |
| 2 | 2.25 [0.65–7.79] | 2.22 [0.64–7.74] | 2.22 [0.72–6.85] | 2.15 [0.69–6.69] | 1.90 [0.54–6.71] | 1.88 [0.53–6.65] | 1.05 0.35–3.18] | 1.03 [0.34–3.14] |
| 3 | 0.64 [0.22–1.88] | 0.59 [0.20–1.76] | 1.94 [0.65–5.81] | 1.84 [0.60–5.64] | 0.72 [0.24–2.21] | 0.72 [0.24–2.21] | 0.86 [0.29–2.56] | 0.84 [0.28–2.54] |
| 4 | 1.18 [0.38–3.67] | 1.09 [0.34–3.44] | 2.22 [0.72–6.85] | 2.09 [0.66–6.64] | 0.62 [0.21–1.89] | 0.62 [0.20–1.89] | 1.72 [0.52–5.69] | 1.63 [0.49–5.46] |
| p trend | 0.462 | 0.480 | 0.189 | 0.229 | 0.430 | 0.432 | 0.694 | 0.715 |

**Notes:**
* Adjusted for; Age. Sex. Physical activity level.
Binary logistic regression.
the propensity for developing cardiometabolic factors encompassing obesity, dyslipidemia, metabolic syndrome, and cardiovascular diseases.

## DISCUSSION

The results of this study provide valuable insights into the relationship between dietary indices and various metabolic and anthropometric parameters in participants with diverse levels of DII, DIL, DGI, and DGL. These findings contribute to the growing body of literature examining the impact of dietary factors on metabolic health, particularly in the context of NAFLD and cardiometabolic disorders. Dietary patterns that heighten insulin levels may potentially worsen hepatic injury by perpetuating chronic inflammation, thereby causing progress towards more severe symptoms of NAFLD (*Valibeygi et al., 2023*). Previous research revealed that consuming diets characterized by elevated overall DGI might amplify markers of systemic inflammation, recognized as pivotal elements contributing to the risk of NAFLD in individuals with T2DM (*Jayedi et al., 2022*; *Popov et al., 2022*). Moreover, a recent study exploring the link between DGI and the onset of NAFLD in individuals with T2DM revealed that a lower DGI was associated with reduced odds of NAFLD development in these patients (*Salavatizadeh et al., 2023*).

The lack of significant differences in demographic characteristics across quartiles of DII, DIL, DGI, and DGL suggests that these dietary indices may not be strongly associated with factors such as age, physical activity level, and BMI in this study population. However, the observed male predominance in higher quartiles of DIL, DGI, and DGL levels warrants further investigation into potential gender-specific dietary patterns and metabolic outcomes. There was no significant difference in distribution of participant demographics between the DII quartiles. Furthermore, no statistically significant variations were noted in factors such as age, BMI, physical activity level, and DIL, DGI, and DGL quartiles. Previous reports have demonstrated similar observations (*Darand et al., 2022*; *Hassanzadeh-Rostami, Ghaedi & Masoumi, 2023*; *Salavatizadeh et al., 2023*). However, a noteworthy finding was that people with higher DIL, DGI, and DGL levels were predominantly men. Similarly, in a previous investigation, it was found that men were in the upper quartiles of DGI and DGL in comparison to women (*Hassanzadeh-Rostami, Ghaedi & Masoumi, 2023*).

The significant differences in body fat percentages across the quartiles of DGI and DGL highlight the potential influence of dietary glycemic index and load on adiposity. Participants in the highest quartiles of DGI and DGL exhibited lower body fat percentages. These findings are inconsistent with previous research conducted as a case-cohort study nested within the European Prospective Investigation into Cancer and Nutrition Study linking high glycemic index diets to increased adiposity and obesity-related outcomes (*Sluijs et al., 2013*). Furthermore, there was a statistically significant difference between the normal waist circumference in the highest and lower quartiles of the DGI. On the other hand, *Salavatizadeh et al. (2023)* found that in T2DM patients with and without NAFLD, waist circumference measures were not significantly different among DGI tertiles. Nevertheless, in a meta-analysis of randomized controlled trials that compared low and

high GI and GL diets in adults with excess weight, no notable variances in fat mass, fat-free mass, or waist circumference were detected (*Perin, Camboim & Lehnen, 2022*).

Contrary to previous studies reporting associations between dietary indices and biochemical parameters, no significant differences were observed in liver enzymes, lipid profiles, and other markers of fatty liver across quartiles of DII, DIL, DGI, and DGL in this study. These findings suggest that factors other than dietary indices, such as genetic predisposition, lifestyle factors, and comorbidities, may play a more significant role in the development and progression of NAFLD.

The lack of association between dietary indices and biochemical parameters underscores the complexity of NAFLD pathogenesis and the need for a multifaceted approach to its management. While dietary interventions, such as reducing overall energy intake and promoting low-glycemic diets, may have potential benefits for liver health, they may not be sufficient on their own to mitigate the risk of NAFLD and cardiometabolic complications. *A prior* study carried out on healthy adults demonstrated that a diet with a high GI resulted in increased levels of liver enzymes, with statistical significance observed solely in men (*Moshtaq et al., 2019*). In a similar vein with the present results, the level of TG did not exhibit an association with DII or DIL in different study populations (*Hassanzadeh-Rostami, Ghaedi & Masoumi, 2023*; *Mozaffari et al., 2019*). On the other hand, some previous results indicated a correlation between elevated DGI and DGL and an increase in serum TG levels in both healthy individuals (*Hosseinpour-Niazi et al., 2013*) and those with T2DM (*Hassanzadeh-Rostami, Ghaedi & Masoumi, 2023*). Discrepancies among studies may arise due to differences in populations studied, study designs, sample sizes, methodological variations, and other pertinent factors.

In recent studies, inverse relationships were observed between DGI and/or DGL and serum HDL-C levels in obese and diabetic subjects (*Hosseinpour-Niazi et al., 2013*; *Lin et al., 2018*). Nevertheless, there was inconsistency in findings concerning the association of both DII and DIL with serum HDL-C, LDL-C and plasma lipid levels (*Mozaffari et al., 2019*; *Nimptsch et al., 2011*). The absence of significant associations between dietary indices and biochemical parameters suggests that other factors or a complex interplay of variables might influence fatty liver and cardiometabolic outcomes. Future research should explore the interplay between dietary factors, genetic susceptibility, and lifestyle behaviors in the development and progression of NAFLD. Longitudinal studies incorporating comprehensive dietary assessments, genetic and metabolic profiling are needed to elucidate the complex mechanisms underlying NAFLD pathogenesis and identify targeted interventions for its prevention and management.

The primary pathogenic factor leading to the development of NAFLD is acknowledged to be a disorder in the metabolism of glucose and lipids (*Samuel & Shulman, 2018*). Significantly reducing overall energy intake, promoting ketosis, or minimizing sugar and carbohydrate consumption might improve liver protection, presenting dietary interventions as a promising strategy for treating NAFLD (*Risi, Tozzi & Watanabe, 2021*). A meta-analysis of randomized clinical trials indicated that diets characterized by a low overall GI may possess anti-inflammatory properties (*Buyken et al., 2014*). Previously, diverse variances in dietary consumption based on levels of DII and DIL have been

documented (*Anjom-Shoae et al., 2020*; *Sadeghi et al., 2020*). Following a comparable pattern, earlier research reported that the highest quartile of DIL demonstrated elevated dietary intake of carbohydrates, fat, protein, dietary fiber, polyunsaturated and saturated fatty acids. The same study reported higher scores of DII were linked to increased consumption of protein, carbohydrates, and dietary fiber, while indicating lower intakes of fat, saturated fatty acids, and polyunsaturated fatty acids (*Anjom-Shoae et al., 2020*).

Similar differences in dietary intake based on DGI and DGL levels have also been documented in earlier research with different study populations (*Lee et al., 2023*; *Sluijs et al., 2013*; *Zhao et al., 2022*). For instance, an earlier study found that dietary protein, saturated fat, monounsaturated and polyunsaturated fatty acids, fiber intakes were higher in the first quartile DGI and DGL than the others (*Zhao et al., 2022*). Similar to the result of the present study that the n-6:n-3 ratio was higher at high DIL levels, another study showed that this ratio was associated with cardiovascular events (*Hayakawa et al., 2012*).

Extensive research spanning numerous decades has scrutinized the impact of fructose on liver-related conditions, obesity, and diabetes. Notably, fructose has been identified as a catalyst for hepatic *de novo* lipogenesis, the accrual of lipids, and the onset of insulin resistance (*Eng & Estall, 2021*). In stark contrast to glucose, the liver predominantly clears fructose from the bloodstream through the GLUT5 transporter, allowing it to circumvent glycolysis—the pivotal step in acetyl-CoA production. Consequently, a substantial volume of acetyl-CoA is expeditiously synthesized subsequent to fructose uptake. While a portion of the acetyl-CoA is utilized for ATP generation within the citric acid cycle, this cycle quickly becomes overburdened, resulting in the rerouting of the surplus acetyl-CoA into *de novo* lipogenesis pathways (*Eng & Estall, 2021*; *Lin et al., 2018*).

The occurrence and rate of incidence for CVD are elevated in patients with NAFLD (*Petroni et al., 2021*). Elevated consumption of carbohydrates, particularly those that increase the GI, has been associated with hyperglycemia and hypertriglyceridemia, influencing the lipid profile and potentially elevating the risk of cardiovascular diseases (*Dwivedi et al., 2022*). Given the significance of hyperinsulinemia in cardiometabolic disorders, identifying shared nutritional factors that significantly contribute to these chronic conditions is of utmost importance. Earlier investigations have indicated a positive correlation between elevated carbohydrate intake and increased body fat as well as dyslipidemia, leading to heightened susceptibility to cardiometabolic disorders (*Papakonstantinou et al., 2017*).

According to our research, those who consumed a diet with a higher DGI were four times more likely to develop MetS than people who followed a diet with a lower DGI (OR, 4.32; 95% CI [1.42–13.11]). In accordance with the present results, meta-analysis of studies employing a dose–response approach showed that higher DGI and DGL was related with a high prevalence of MetS (*Askari et al., 2021*; *Zhang et al., 2020*).

Odds ratios for Q2 (OR, 0.22; 95% CI [0.06–0.81]), Q3 (OR, 0.23; 95% CI [0.06–0.85]), and Q4 (OR, 0.25; 95% CI [0.07–0.95]) showed relationships between DGI and obesity in the present study. A recent previous study showed an association between DGL and an increased likelihood of obesity in women, while no such association was found in men among participants with atherosclerosis (*Behbahani et al., 2023*).

The variations observed between the present findings and those of previous research may stem from several factors inherent to the methodology and population characteristics employed. Variations in dietary assessment tools, ranging from food frequency questionnaires to dietary recall methods, could contribute to divergent results. Additionally, the age distribution of the study populations may introduce heterogeneity in the observed associations. Furthermore, the inclusion of different variables as confounding factors, such as socioeconomic status, physical activity level, and comorbidities, across studies may further obscure comparability. Methodological nuances in the computation of dietary indices, including DII, DIL, DGI, and DGL scores could also contribute to discrepancies in findings. It is noteworthy that the existing literature exhibits a paucity of studies specifically investigating the role of DII, DIL, DGI, and DGL in patients diagnosed with NAFLD, thereby underscoring the need for further research in this domain to elucidate their potential implications in this patient population.

### Strengths and limitations

This study has strengths and limitations that need to be addressed. The primary strength of this study lies in the first investigation for the association of DIL, DII, DGI, and DGL with cardiometabolic risk factors in patients with NAFLD. Furthermore, analyses for various potential confounding factors were adjusted to establish an independent association. Controlling for confounders strengthens the validity and reliability of our results. On the other hand, one notable limitation of this study is its cross-sectional design, which inherently restricts our ability to infer causality between dietary factors and cardiometabolic risk factors in NAFLD. While we observed associations between dietary indices and health outcomes, longitudinal studies are needed to confirm causality. Additionally, using 24-h food records for assessing dietary intake may introduce limitations such as recall bias and underreporting.

## CONCLUSIONS

This study has provided initial evidence regarding the association of DIL, DII, DGI, and DGL with cardiometabolic risk factors in patients with NAFLD. The results of this study revealed that diet with a high glycemic index is linked to an increased odd of having MetS in individuals with NAFLD. Notably, the dietary energy and nutrient intake of participants consuming diets with different levels of DIL, DII, DGI, and DGL showed variations. Although comprehending the connections of dietary factors on disease prognosis is one of the important issues in the management of NAFLD, present study outcomes suggest that low glycemic and insulinemic dietary composition may provide benefits management or prevention of MetS. Considering the preliminary nature of the current findings, further studies, particularly those employing a prospective dietary interventional or observational design, high-quality cohort studies and clinical trials, are needed to provide additional insights.

# ACKNOWLEDGEMENTS

The authors would like to thank the staff of Hacettepe University Adult Hospital, Department of Gastroenterology and Radiology.

## Funding

The authors received no funding for this work.

## Competing Interests

The authors declare that they have no competing interests.

## Author Contributions

- Gulsum Gizem Topal conceived and designed the experiments, performed the experiments, analyzed the data, prepared figures and/or tables, authored or reviewed drafts of the article, and approved the final draft.
- Sumeyra Sevim analyzed the data, prepared figures and/or tables, authored or reviewed drafts of the article, and approved the final draft.
- Damla Gumus analyzed the data, prepared figures and/or tables, authored or reviewed drafts of the article, and approved the final draft.
- Hatice Yasemin Balaban conceived and designed the experiments, authored or reviewed drafts of the article, and approved the final draft.
- Muşturay Karçaaltıncaba conceived and designed the experiments, authored or reviewed drafts of the article, and approved the final draft.
- Mevlude Kizil conceived and designed the experiments, authored or reviewed drafts of the article, and approved the final draft.

## Human Ethics

The following information was supplied relating to ethical approvals (*i.e.*, approving body and any reference numbers):

The research received approval from the Hacettepe University Health Sciences Research Ethics Committee (GO 22/1045).

## Data Availability

The raw data is available in the Supplemental File.

## Supplemental Information

Supplemental information for this article can be found online at http://dx.doi.org/10.7717/peerj.17810#supplemental-information.

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
