# Peer review of "Are dietary factors associated with cardiometabolic risk factors in patients with non-alcoholic fatty liver disease?"

_PeerJ, doi:10.7717/peerj.17810_

## Round 0.1 · original submission · Major Revisions

Dear Dr. Kizil,

I have now received the reviews of your paper. As you can see, both reviewers especially reviewer 1, provide detailed comments to improve your paper and strengthen the arguments. Please revise the paper accordingly.

sincerely,
Shaw Badenhorst

·

Basic reporting

The authors have done a cross-sectional study to clarify the association between dietary factors with cardiometabolic risk factors in patients with non-alcoholic fatty liver disease. They are minor grammatical errors in the manuscript so extra review of the text can be done. In the following section, some comments are given to improve the quality of the study.
Modifying the abstract section
1. Specify a separate title for the conclusion in the abstract.
2. It is suggested to use more numbers in expressing the results related to the abstract section.
Improvement of the introduction section
3. In order to better express the prevalence of non-alcoholic fatty liver disease, it is better to state the percentage of its prevalence in the world and developing and advanced countries.
4. It is suggested to make a better comparison between the present study and similar previous studies and to state its strengths and limitations.
5. It is suggested to summarize the mechanisms mentioned in the introduction and give more explanations in the discussion section.

Experimental design

The transparency of the method
6. What do you mean by the phrase "took medications or supplements that could influence physical functioning"?
7. Wasn't it necessary to consider a body mass index range for patients to enter the study?
8. Have other chronic diseases been considered?
Further description of the results
9. There is no description of the gender distribution of the participants in the Demographic characteristics of the participants’ section.
10. Considering the effect of male and female hormonal differences on the results, please express the results separately for both sexes.
11. State the cutoff used for each dietary pattern. It should also be stated in the upper part of the tables.
12. In table 1, 2, for each variable, state its values in the entire population.

Validity of the findings

Improve the discussion section
13. In the discussion section, it is appropriate to explain about the articles that are aligned and not aligned with our results, but the strengths and limitations of these studies and the reason for the difference or similarity between the results of these studies and our study should be stated.

Reviewer 2 ·

Basic reporting

See comments on pdf

Experimental design

See comments on pdf

Validity of the findings

See comments on pdf

Additional comments

See comments on pdf

Annotated reviews are not available for download in order to protect the identity of reviewers who chose to remain anonymous.

---

## Round 0.2 · accepted · Accept

Dear Dr. Kizil,

I have the pleasure to inform you that the two reviewers are satisfied that you adequately addressed their concerns in the revised paper, and that your paper can now be accepted by PeerJ for publication.

Sincerely,
Dr. Shaw Badenhorst

·

Basic reporting

The authors have done everything well. This article is suitable for publication.

Experimental design

The authors have done everything well. This article is suitable for publication.

Validity of the findings

The authors have done everything well. This article is suitable for publication.

Additional comments

The authors have done everything well. This article is suitable for publication.

Reviewer 2 ·

Basic reporting

Accept

Experimental design

Accept

Validity of the findings

Accept

Additional comments

Accept